# An Innovative Ultrasound Technique for Early Detection of Kidney Dysfunction: Superb Microvascular Imaging as a Reference Standard

**DOI:** 10.3390/jcm11040925

**Published:** 2022-02-10

**Authors:** Zaher Armaly, Munai Abu-Rahme, Safa Kinaneh, Basem Hijazi, Nayef Habbasshi, Suheil Artul

**Affiliations:** 1Department of Nephrology, Nazareth Hospital-EMMS, Nazareth and the Azrieli Faculty of Medicine, Bar Ilan University, Zafed 1330311, Israel; munai.abu@gmail.com (M.A.-R.); safakinaneh@gmail.com (S.K.); 2Administration Department, Azrieli Faculty of Medicine, Bar Ilan University, Zafed 1330311, Israel; basem.hijazi@biu.ac.il; 3Department of Nephrology, HaEmeq Hospital, Afula 1804309, Israel; habashinayaf@gmail.com; 4Department of Radiology, Nazareth Hospital-EMMS, Nazareth and the Azrieli Faculty of Medicine, Bar Ilan University, Zafed 1330311, Israel; suheilartul@nazhosp.com

**Keywords:** chronic kidney disease, superb microvascular imaging, serum creatinine, kidney fibrosis, ultrasound, renal biopsy

## Abstract

Background: Superb microvascular imaging (SMI) is an innovative ultrasound image processing technique that provides greater detail and better visualization of small branching vessels. We assume that SMI will provide sufficient information regarding the severity of chronic kidney disease (CKD) and reflecting histological changes. Aims: The aims was to assess the capabilities of SMI imaging regarding the early detection of kidney dysfunction and renal fibrosis in comparison to the reference standard renal biopsy for the early diagnosis of kidney fibrosis. Methods: SMI was performed in patients (*n* = 52) with CKD stage 2–5, where some of them underwent biopsy proven CKD and fibrosis as part of the diagnosis. In addition, biochemical tests were performed, including kidney function tests, urine collection for proteinuria, and the estimation of GFR by MDRD or CKD-EPI eGFR in CKD patients and healthy controls (*n* = 17). All subjects underwent SMI, where vascularity is expressed as the SMI index (a low index reflects low vascularity/fibrosis and vice versa). Results: The SMI vascular index was significantly lower in CKD patients as compared with healthy controls (72.2 ± 12.9 vs. 49.9 ± 16.7%, *p* < 0.01). Notably, a moderate correlation between the SMI index and eGFR was found among the CKD patients (r = 0.56, *p* < 0.001). Similarly, a strong correlation was found between SCr and the SMI index of the diseased subjects (r = −0.54, *p* < 0.001). In patients who underwent renal biopsy, the SMI index corresponded with the histological alterations and CKD staging. Conclusions: This study demonstrated that SMI imaging may be utilized in CKD patients of various stages for the evaluation of chronic renal morphological changes and for differentiation between CKD grades.

## 1. Introduction

Chronic kidney disease (CKD) is characterized by a decline in glomerular filtration rate (GFR), a progressive increase in proteinuria, elevated blood pressure, and high risk of advanced kidney failure that requires replacement therapy. In addition, advanced CKD is characterized by morphologic renal changes that include varying degrees of fibrosis [1,2]. For several decades, the evaluation of renal disorders has been based on a variety of conventional methods, including biochemical analysis, ultrasound, computerized tomography (CT), and magnetic resonance imaging (MRI). Although the progression of CKD can be assessed using routine biochemical tests, such as the proteinuria and serum creatinine levels, assessment of the grade of renal fibrosis requires renal biopsy.

Renal biopsy plays an important role in the establishment of an exact diagnosis and in determining the degree of active and chronic microscopic changes regardless of the initial cause of the ongoing kidney disease. Thus, renal biopsy is an essential process which helps to assess the severity, prognosis, and treatment of certain renal illnesses. As an invasive medical procedure, biopsy is performed with the assistance of ultrasound or CT scanning. Usually, patients who have had renal biopsy will be observed for at least 24 h before being discharged. Complications such as bleeding, arteriovenous fistula, pain, chronic hypertension, renal insufficiency, and infection may occur at low incidences.

Conventional renal ultrasound may reveal reduced renal length and cortical thickness, as well as increased cortical echogenicity, which may suggest the presence of chronic, atrophic morphological changes in a variety of renal diseases. However, as opposed to renal length and cortical thickness, cortical echogenicity is not measurable. Moreover, in diabetic patients, kidneys frequently do not show atrophic change via conventional renal ultrasound, even in advanced diabetic nephropathy (DN) with CKD grades 3 and 4. Therefore, conventional renal ultrasound is, generally, not informative for the evaluation of the progression of DN and not useful for the differentiation between the CKD grades [1,2,3].

Hassan et al. found that in patients with advanced DN, shear wave elastography (SWE) imaging may be utilized as a simple and practical method for the quantitative evaluation of chronic morphological changes and for differentiation between CKD grades [4]. Thus, SWE based on acoustic radiation force impulse (ARFI) technology is a noninvasive and cost-effective ultrasound diagnostic method that has been developed for the evaluation of tissue elasticity [3,5,6,7,8]. Essentially, SWE technology measures the velocity of the shear wave (in m/s) that passes through the examined tissue and then converts this measurement into a tissue stiffness measurement that is expressed in kPa [3,5,6,7,8,9]. A number of recent applications of SWE have focused on the evaluation of the mechanical properties of renal tissue [3,5,6,7,8,9]. However, SWE has few operational and reliability limitations, especially in obese patients. Specifically, SWE does not accurately reflect the degree of fibrosis as biopsy analysis does. Xiao et al. determined the best method for diagnosing liver fibrosis (LF) in nonalcoholic fatty liver disease (NAFLD) by applying biochemical tests such as the aspartate aminotransferase to platelets ratio index (APRI), fibrosis-4 index (FIB-4), BARD score, and NAFLD fibrosis score (NFS), and imaging techniques such as FibroScan, SWE, and magnetic resonance elastography (MRE) [10]. These authors found a higher sensitivity of MRE over the other examined diagnostic tools [10]. Therefore, there is a need for a non-invasive imaging methodology that can provide comprehensive and in-depth qualitative and quantitative assessments of tissue elasticity and the severity of histological changes. One of the emerging technologies that can overcome these limitations is superb microvascular imaging (SMI). Superb microvascular ultrasound imaging is an innovative vascular imaging technique that allows for the detection of small vessels with low-velocity blood flows with high frame rates, less motion artifact, and high resolution without the use of contrast agent [11,12,13,14,15,16,17,18,19,20,21,22,23]. Unfortunately, so far, SMI has not been utilized yet to estimate renal fibrosis in patients with CKD. Therefore, in the current study, we aimed to investigate whether renal SMI imaging can serve as a practical and noninvasive tool for the quantitative assessment of kidney dysfunction and renal fibrosis in CKD patients at stages 2–5, where conventional renal ultrasound is not informative for the evaluation of the progression of CKD and not useful for differentiation between CKD grades. Therefore, the current study assessed the capabilities of SMI regarding the early detection of kidney dysfunction and renal fibrosis in comparison to the reference standard renal biopsy, in order to determine the usefulness of SMI in the early diagnosis of kidney fibrosis even without major changes in SCr.

## 2. Materials and Methods

### 2.1. Study Design

This was a prospective, open-label analytical study. The participants in this study were patients who regularly visit the out-patient Nephrology Clinic to resume their nephrological follow-up, including the EMMS outpatients’ clinic. The study was approved by the Nazareth Hospital EMMS Human Research Review Committee and carried out at Nazareth Hospital. Informed consent was obtained from all subjects involved in the study. All patients received a proper explanation regarding the SMI test and were asked for their approval to participate in the study (signed consent form). Patients with available biopsy were recruited from HaEmek Medical Center. The demographics (age, gender, background diseases, BMI, duration of the disease, and medication) and laboratory data of the studied patients were collected from their medical records and placed in an appropriate Excel table. These data also included the latest routine complete blood count, biochemical analyses, and biopsy results when available. Appropriate patients who met the inclusion criteria underwent SMI no later than 6 months from biopsy.

This study included 52 patients (27 males and 25 females) and 17 controls (11 males and 6 females). Background diseases among CKD patients included diabetes mellitus (Type 2) (*n* = 16), hypertension (*n* = 17), hyperlipidemia *n* = (14), and heart disease (*n* = 4). All patients and normal health controls underwent SMI of the kidney as described above, carried out by the same senior physician (Dr. Artul, head of the Department of Radiology—EMMS Hospital, with more than 20 years of experience). The clearest image of the region of interest (ROI) was chosen by Dr. Artul and analyzed accordingly. The selected ROIs represented the generalized vascular obtained mapping of the kidney cortex. Moreover, the analysis of the images was performed by the same physician.

### 2.2. Superb Microvascular Ultrasound Imaging

Toshiba’s AplioTM 500 ultrasound system provides a full suite of tools for evaluating vessels and hemodynamics, including color Doppler Imaging, power Doppler, Advanced Dynamic FlowTM and pulsed wave Doppler. Combined with the contrast capability of the system, this provides an outstanding tool-kit for vascular diagnosis [24,25]. To expand the range of visible flow in ultrasound, Toshiba’s Aplio 500 is now equipped with a new technology, specifically for imaging very low-flow states. This new feature called superb microvascular imaging (SMI) is an innovative ultrasound Doppler technique. SMI is a unique ultrasound Doppler technique that employs a special algorithm that allows for the visualization of minute vessels with slow velocity without having to use a contrast agent [24,25]. The advantages of SMI are: 1. Low-velocity flow visualization; 2. high resolution; 3. minimal motion artefact; 4. high frame rates. SMI applies a powerful algorithm that effectively separates flow signals from overlying tissue motion artifact, and this suppresses clutter and reveals the true blood flow, providing a perspicuous image [24,25]. SMI is available in two modes, the color SMI (cSMI)—a color informational mode—and grayscale mode—the monochrome mode (mSMI)—which improve the sensitivity by subtracting the background information [24,25].

Impressions give us information about the quality of the kidney vascularity; they are described by three stages: 1 represents good vascularity, 2 represents intermediate vascularity, and 3 represents poor vascularity.

Shape gives us information about the anatomy of the kidney vessels and their integrity. It is described by three stages: 1 represents a normal vessel, 2 represents a slightly distorted vessel, and 3 represents a very distorted vessel. It should be emphasized that the terms “impression” and “shape” used in this context are not imaging parameters; rather, they are operator estimations.

### 2.3. Subjects and Parameters

–Inclusion Criteria:
Patients with CKD at stages 2–5 (pre-dialytic).Patients with known CKD, abnormal serum creatinine, and proteinuria levels even when GFR apparently normal.At age ≥18 y.Exclusion Criteria:
Acute kidney injury.Pregnant women.Patients with severe liver diseases.Inter-current illness such as fever or sepsis.Allergic rhinitis.Hydro-nephrosis.

Patients diagnosed as suffering from chronic kidney diseases at CKD stages 2–5 and confirmed by the modification of diet in renal disease (MDRD) equation or CKD-EPI were recruited at the Department of Nephrology, Nazareth Hospital between 2017 and 2019 after a signing consent form.

All patients underwent blood and urine tests to determine kidney function according to creatinine and BUN levels. The presence and severity of proteinuria was determined either by total urinary protein excretion or the albumin to creatinine ratio. Only patients who exhibited kidney dysfunction were included in the study and underwent single SMI. In the cases where renal biopsy was available, we used the obtained etiology and renal histological changes to compare them with our findings via SMI. It should be emphasized that the radiologist was blinded to the kidney function/histological grading of the patients undergoing the imaging procedure.

Only patients who underwent kidney biopsy in the last 3–6 months were included in the comparison between the sensitivity of SMI and biopsy findings. Regarding patients who did not undergo renal biopsy, their blood chemical analysis for SCr and BUN, besides 24-urine collection for GFR (CCT), served as the gold standard for staging renal failure degree.

Healthy volunteers with normal kidney function (*n* = 17) served as a control group.

For each patient, the severity of fibrosis was presented as the percentage of renal tissue; SMI, the impression, shape, and size of the kidney were obtained. In addition, renal blood flow and mean vascular score (stiffness, perfusion, and mean bifurcation angle) were determined.

The obtained results were analyzed for possible correlation with serum creatinine, proteinuria, and GFR. The severity of renal disruption observed via SMI was compared to the degree of fibrosis in renal biopsy when available.

### 2.4. Statistical Analysis

The quantity-collected data are exhibited as mean ± SD, median, and range. Qualitative data were described as frequencies and percentages. The statistical differences between SMI and renal biopsy groups were assessed using the Mann–Whitney U test. Correlations between ordinal variables/ordinal and quantity data were determined by applying Spearman’s correlation coefficient test. Correlations between qualitative data were examined via the Chi square test or Fisher’s exact test, while the tests were chosen according to pre-assumptions which were required before using those tests. A *p*-value ≤0.05 indicated statistically significant changes.

## 3. Results

### 3.1. Description of Main Data of the Participants

Sixty-nine subjects were included in the current study. Subjects were divided into two groups; CKD patients and healthy controls. The first group included 52 participants and the second contained 17 participants; 52% were men and 48% were women in total, with a mean age of 56.8 ± 17.2 years in patients and 36.1 ± 12.3 years in controls. In the CKD group, 11 subjects had CKD stage 1, 13 had CKD stage 2, 23 had CKD stage 3, 3 had CKD stage 4, and 1 person had CKD stage 5; 1 had insufficient collected data to determine the CKD stage. In SMI, 63.8% had good impressions and 64.7% normal-shaped kidneys, 23.2% had intermediate impressions and 25.0% had slightly distorted shapes, 13.0% had poor impressions, and 10.3% had very distorted shapes (Table 1).

### 3.2. Kidney Function Parameters and SMI Evaluations in All Subjects

The values of studied parameters for all participants in this study are summarized in Table 2. Serum creatinine values ranged between 0.49 and 3.5 mg/dL. The mean average of all subjects was 1.22 ± 0.57 mg/dL. In total, 25% had creatinine value less than 0.80 mg/dL and 75% had less than 1.49 mg/dL; the median value was 1.13 mg/dL. The mean eGFR was 73.75 ± 34.49 mL/min/1.73 m² and 73.29 ± 34.43 mL/min/1.73 m² by MDRD and CKD-EPI, respectively (normal range is ≥ 90 mL/min/1.73 m²). In total, 25% had an MDRD eGFR value less than 43.93 mL/min/1.73 m² and 75% had less than 105.90 mL/min/1.73 m². The median, minimal, and maximal values of MDRD eGFR were 64.12 mL/min/1.73 m², 14.07 mL/min/1.73 m², and 158.51 mL/min/1.73 m², respectively (Table 2). A total of 25% had CKD-EPI eGFR values less than 43.30 mL/min/1.73 m², and 75% had less than 107.00 mL/min/1.73 m. The median, minimal, and maximal values of CKD-EPI eGFR were 65.50 mL/min/1.73 m², 13.30 mL/min/1.73 m², and 133.60 mL/min/1.73 m², respectively (Table 2). The mean value of proteinuria in CKD subjects was 1.74 ± 1.55 gr/24 h (normal values = 0–0.020 gr/24 h). In total, 25% of participants had no protein in their urine, and 75% had less than 2.3 gr/24 h. The median value was 1.54 gr/24 h, the minimal value was 0, and the maximal was 6.6 gr/24 h (Table 2). In some patients (*n* = 13), the albumin/creatinine ratio (ACR) was determined, where the mean value of ACR was 325 ± 593 mg/g (normal value of ACR is <30 mg/mL). In total, 25% of participants had less than 14 mg/g in their urine, and 75% had less than 331 mg/g. The median, minimal, and maximal values were 39 mg/g, 5 mg/g, and 1785 mg/g, respectively (Table 2).

Considering the radiologic parameters, the mean vascular index was 55.39 ± 18.5% (normal values are >68.63); 25% of participants had a vascular index less than 46.50%, and 75% had less than 70.00%. The median, minimal, and maximal values of the vascular index were 55.00%, 20.00%, and 90.00%, respectively (Table 2). The mean value of kidney size was 11.19 ± 1.52 cm (normal value ≤ 12 cm); 25% of participants had a kidney size smaller than 10.20 cm, and 75% had a size smaller than 12.00 cm. The median, minimal, and maximal values of kidney size were 11.20 cm, 8.30 cm, and 17.20 cm, respectively (Table 2).

### 3.3. Comparison between Control and CKD Patients in Main Tested Parameters

**Controls:** In the control group, the mean serum creatinine was 0.78 ± 0.11 mg/dL (normal values range between 0.7 and 1.3 mg/dL). In total, 25% of this group had serum creatinine less than 0.74 mg/dL, and 75% had less than 0.84 mg/dL; the median was 0.80 mg/dL, as shown in Table 3 and Figure 1A. The mean eGFR was 113.44 ± 16.59 mL/min/1.73 m² and 108.32 ± 21.66 mL/min/1.73 m² by MDRD and CKD-EPI in the control group, respectively; (normal range is ≥90 mL/min/1.73 m²). In total, 25% had an MDRD eGFR value less than 105.90 mL/min/1.73 m², and 75% had less than 117.61 mL/min/1.73 m². The median value of MDRD eGFR in the control group was 110.40 mL/min/1.73 m² (Table 3 and Figure 1B). In total, 25% had a CKD-EPI eGFR value less than 99.50 mL/min/1.73 m², and 75% had less than 119.10 mL/min/1.73 m. The median value of CKD-EPI eGFR was 115.00 mL/min/1.73 m² (Table 3). Proteinuria and ACR levels are not demonstrated in the healthy control group.

Considering the radiologic parameters, the mean vascular index was 72.16 ± 12.92% (normal values > 68.63%); 25% of participants had a vascular index less than 60.00%, and 75% had less than 82.00%, The median value of the vascular index was 74.00 (Table 3 and Figure 1C). The mean value of kidney size was 11.19 ± 0.9 cm (normal value ≤ 12 cm); 25% of healthy controls had a kidney size smaller than 11 cm, and 75% had a size smaller than 11.80 cm. The median value of kidney size was 11.20 cm (Table 3).

**Patients:** In the CKD patients group, the mean serum creatinine was 1.33 ± 0.59 mg/dL; 25% of this group had serum creatinine less than 0.88 mg/dL, and 75% had less than 1.62 mg/dL; the median was 1.27 mg/dL, as shown in Table 3 and Figure 1A. **The** mean eGFR was 63.06 ± 29.95 mL/min/1.73 m² and 62.79 ± 30.43 mL/min/1.73 m² by MDRD and CKD-EPI, respectively; (normal range is ≥90 mL/min/1.73 m²). In total, 25% had an MDRD eGFR value less than 41.01 mL/min/1.73 m², and 75% had less than 85.79 mL/min/1.73 m². The median value of MDRD eGFR was 55.04 mL/min/1.73 m² (Table 3 and Figure 1B). A total of 25% had a CKD-EPI eGFR value less than 40.50 mL/min/1.73 m², and 75% had less than 93.30 mL/min/1.73 m^2^. The median value of CKD-EPI eGFR was 53.60 mL/min/1.73 m² (Table 3).

Considering the radiologic parameters, the mean vascular index was 49.91 ± 16.72% (normal values > 68.63); 25% of participants had a vascular index less than 33.75%, and 75% had less than 61.90%, The median value of vascular index was 50% (Table 3 and Figure 1C).

The mean value of kidney size was 11.19 ± 1.69 cm (normal values ≤ 12 cm); 25% of CKD subjects had a size value less than 10.05 cm, and 75% had a value less than 12.00 cm. The median value of kidney size was 11.00 cm (Table 3).

### 3.4. Linear Correlation between All Main Tested Parameters in All Participants

Correlation analysis was performed for all parameters, and the results are shown in Table 4 and Figure 2 and Figure 3. A moderate linear correlation was observed between the Scr/eGFR and the radiological parameters (index, impression, and shape), which means that high values of Scr correspond with abnormal radiological parameters: vascular index (r = −0.60), impression (r = 0.65), and shape (r = 0.65) (Figure 2A,B). A weak linear correlation was observed between proteinuria and the vascular index (−0.251) (Figure 3), and this correlation was moderate regarding impression (r = 0.309) and shape (r = 0.455) (Table 4).

As expected, a strong linear correlation was observed between eGFR and Scr (r = −0.939; −0.904) (Table 4 and Figure 2A,B). The vascular index was given by the SMI screen output along the radiologist impression and kidney shape. As shown in Table 4, a strong linear correlation between the vascular index and the shape and impression, which supports the validity of the SMI in detecting suspected kidney damage, was spotted. In addition, a strong correlation between the vascular index and Scr/eGFR was observed (Figure 2C,D).

### 3.5. Linear Correlation between All Main Tested Parameters in CKD Patients

Correlation analysis was performed in CKD patients, and the results are shown in Table 5 and Figure 2. A moderate linear correlation was observed between the Scr and the radiological parameters index (r = −0.54), impression (r = 0.57), and shape (r = 0.58). A strong linear correlation was observed between eGFR and SCr (r = −0.953) (Table 5 and Figure 2A,B). As can be seen in Table 5 and Figure 2C,D, a strong linear correlation between the vascular index, shape, and impression was found (r = −0.853; −0.87, respectively). Likewise, a strong negative correlation was found between CKD stage and SMI (Figure 3A); specifically, the more advanced the CKD stage, the lower the SMI. In contrast, a weak linear correlation was observed between proteinuria and the vascular index (r = −0.251) (Table 5 and Figure 3B,C). Figure 3C illustrates the correlation between the SMI index of 11 biopsy-approved CKD patients and renal fibrosis. As proven in biopsy, the higher the fibrosis, the lower the SMI vascular index that was obtained (Figure 3D).

Figure 4 illustrates the SMI screen tests and shows the anatomical arcuate arteries in a healthy control with the impression of good vascularity; Figure 4B, illustrates the SMI Doppler in a healthy subject with eGFR = 110 mL/min, vascular index = 65.7%, and normal arcuate arteries; Figure 4C illustrates the SMI Doppler in CKD Stage 3a patients with eGFR = 44 mL/min, vascular index = 41.7%, and slightly distorted arcuate arteries with intermediate vascularity; Figure 4D illustrates the SMI Doppler in CKD stage 3b patients with eGFR = 40 mL/min, vascular index = 26.7%, and very distorted arcuate arteries with poor vascularity.

## 4. Discussion

Superb microvascular imaging (SMI) is an innovative ultrasound image processing technique that shows greater detail and better visualization of small branching vessels by using a unique algorithm that offers high frame rates, less clutter, and less tissue motion artifact, which could not previously be achieved without the use of contrast agent [24]. Renal circulation has unique anatomical and functional characteristics. The microvascular blood flow in the kidney is generally regulated according to the specific needs of the tissues as long as the arterial pressure is sufficient to sustain adequate tissue perfusion [26,27]. One important factor affecting kidney function is the status of blood supply to kidney glomerulus via the interlobular arteries or cortical radial arteries. These vessels are directed toward the cortical substance to form end-arteries through lateral branches as afferent arterioles to supply renal corpuscles, enter Bowman’s capsule, and end in the glomerulus. The size of microvasculature in the kidney is very small (<200 μm in diameter) [27,28]. The hemodynamic status in kidney cortical microvasculature is associated with kidney function [29,30]. The assessment of cortical microvasculature blood flow representing kidney perfusion is challenging due to the small caliber of the vessels with slow flow.

The present study evaluated the usefulness of SMI in the diagnosis of kidney fibrosis. Specifically, we assessed the capabilities of superb microvascular ultrasound imaging for the early detection of renal fibrosis in comparison to the reference standard renal biopsy in order to determine the usefulness of SMI in the early diagnosis of kidney fibrosis even without major changes in SCr. We clearly demonstrated that the SMI index was significantly lower in CKD patients as compared with healthy controls (72.2 ± 12.9 vs. 49.9 ± 16.7%, *p* < 0.01). Furthermore, a strong correlation between the SMI index and eGFR was found among the CKD patients. Similarly, a strong correlation between the SCr and SMI index of the diseased subjects was found. This correlation is basic, and Scr is an indicator of the kidney’s function; as for the SMI, this fact empowers its reliability and the ability to demonstrate when there is a decrease in the kidney clearance function. In addition, a weak linear correlation was observed between proteinuria and the vascular index, which could be explained by the fact that not all diabetes mellitus patients demonstrate proteinuria in the early years of the disease following the diagnosis or exhibit a nonalbuminuric progressive renal impairment in both type 1 and 2 DM [31,32]. A moderate linear correlation was observed between the Scr/eGFR and the radiological parameters (index, impression, and shape), which means that high values of Scr correspond with abnormal radiological parameters: SMI. Among those who underwent kidney biopsy, SMI reflects the fibrotic alterations as well as CKD staging. These findings suggest that in patients with CKD of various stages, SMI imaging may be utilized as a simple and practical method for the evaluation of chronic kidney dysfunction mainly in the early stages of CKD, as well as the degree of fibrotic changes associated with this clinical setting. Collectively, the most remarkable finding of the current study is the differences in imaging parameters between healthy and CKD populations. As expected, there are modest correlations when looking at CKD population concerning blood parameters which correlate well with each other, with radiologic parameters, except size, correlating well with each other; this is a new yet not surprising finding, but there are only modest correlations between these two broader categories (blood and imaging). Urine parameters were not found to correlate well with SMI. These modest correlations are meaningful in a technique that is new and has not been explored in the nephrological field. The correlations may be improved upon by increasing the sample size.

### Study Limitation

Despite the encouraging results of the current study concerning the validity of SMI imaging in detecting early CKD and adverse kidney fibrotic alterations, there are limitations: 1: the numbers of patients and controls were low; 2: some of the healthy controls had slightly lower SMI than the normal limit, which may stem either from the nature of the applied US or the presence of minor histological/vascular changes in the kidney despite normal SCr/GFR; 3: the age of the healthy subjects was not comparable to that of patients; 4: not all patients underwent kidney biopsy. These limitations may be overcome by the mobilization of more CKD patients of various severities and matched healthy subjects.

In summary, the most remarkable finding of the current study is the differences in imaging parameters between healthy and CKD subjects. Specifically, this study demonstrates that in patients with CKD of various stages, SMI imaging may be utilized as a simple and practical method for the evaluation of the chronic renal morphological changes and for the differentiation between CKD grades, as we managed to demonstrate how SMI reflects the kidney functional status by translating it to percentages of the vascularity index. In other words, a shorter distance of the cortical end vessel to the kidney capsule indicates the higher sensitivity of SMI in depicting smaller vessels in the cortex near the kidney capsule [26], and also to a classification of the shape and impression to three stages according to each patient’s kidney.

## 5. Conclusions

The main contribution of our study is the provision of a noninvasive approach to detect chronic kidney dysfunction and fibrosis, as well as the potential replacement of kidney biopsy as a diagnostic tool for this disease state. Despite remarkable advances in medicine, the detection of kidney dysfunction and fibrosis may be delayed due to the limited accuracy of SCr at early stages of the disease. Moreover, not all patients undergo kidney biopsy, and the available noninvasive images are insensitive. Early detection might help to initiate an appropriate treatment and to prevent progress in kidney fibrosis, with a possible reduction in morbidity, mortality and medical expenditure, including decreasing the need for acute and chronic dialysis. In summary, SMI will provide sufficient information regarding the severity of chronic kidney disease and reflecting fibrotic changes.

## Figures and Tables

**Figure 1 jcm-11-00925-f001:**
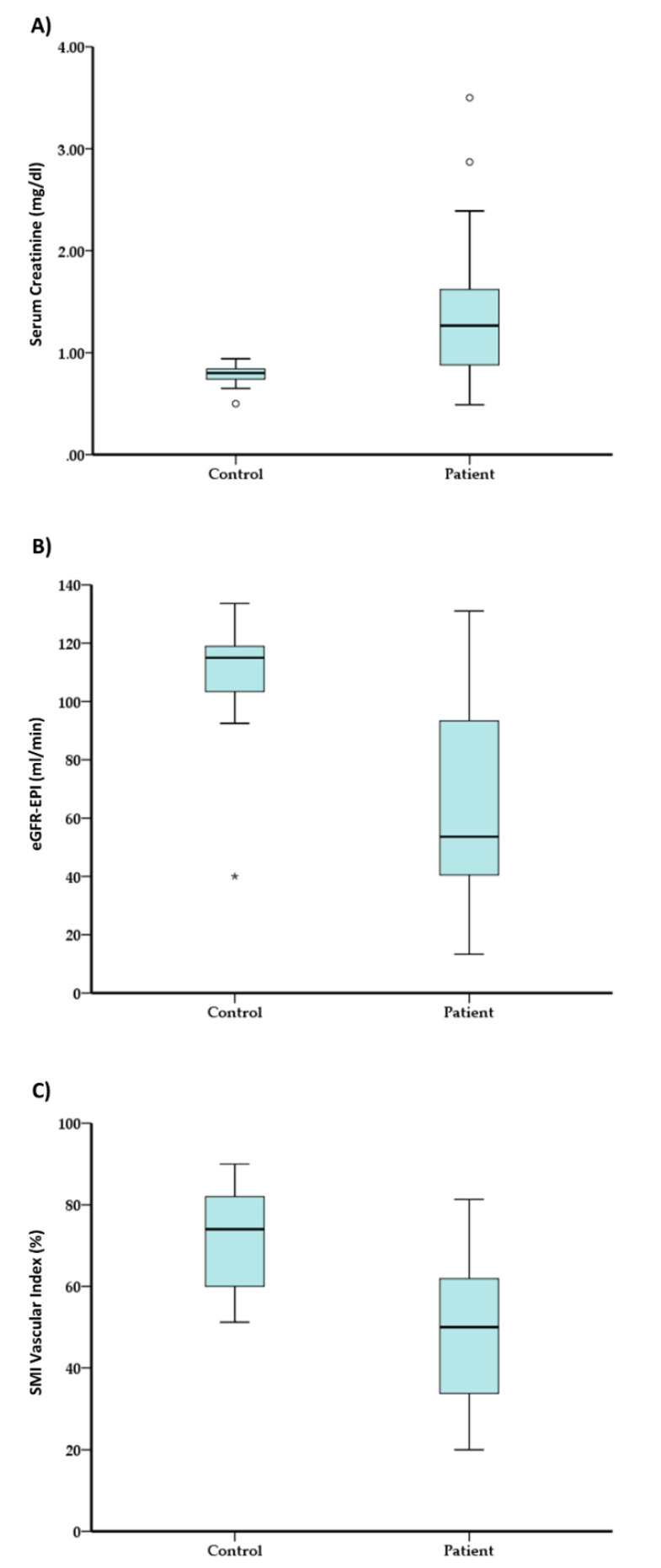
(**A**) Mean serum creatinine in both control and CKD groups; (**B**) mean CKD-EPI eGFR-EPI in control and CKD groups; (**C**) SMI index in healthy subjects and CKD patients. (*) and (°) marks the outlier values.

**Figure 2 jcm-11-00925-f002:**
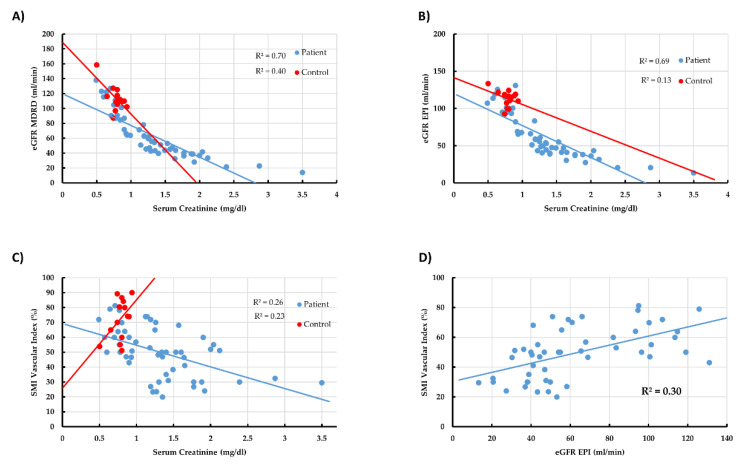
(**A**) Correlation between eGFR and serum creatinine in CKD patients (red), and control group (blue); (**B**) correlation between CKD-EPI eGFR-EPI index and serum creatinine in patients and healthy controls; (**C**) correlation between vascularity index and serum creatinine in patients and healthy controls; (**D**) correlation between vascular index and eGFR-EPI in patients and healthy controls.

**Figure 3 jcm-11-00925-f003:**
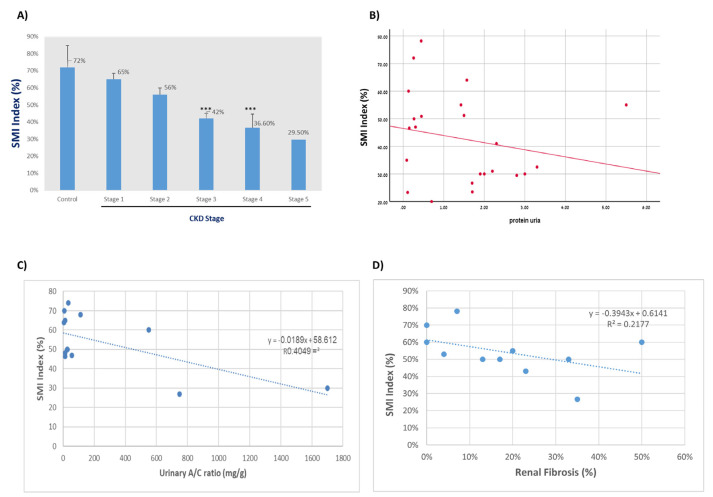
Correlation between vascular index and (**A**) CKD stages (*** *p* < 0.001 as compared to stage); (**B**) Proteinuria (mg/24 h) in 22 CKD patients; (**C**) urinary albumin/urinary creatinine ratio (mg/g.) in 13 CKD patients; (**D**) fibrosis in 6 biopsy-approved CKD patients.

**Figure 4 jcm-11-00925-f004:**
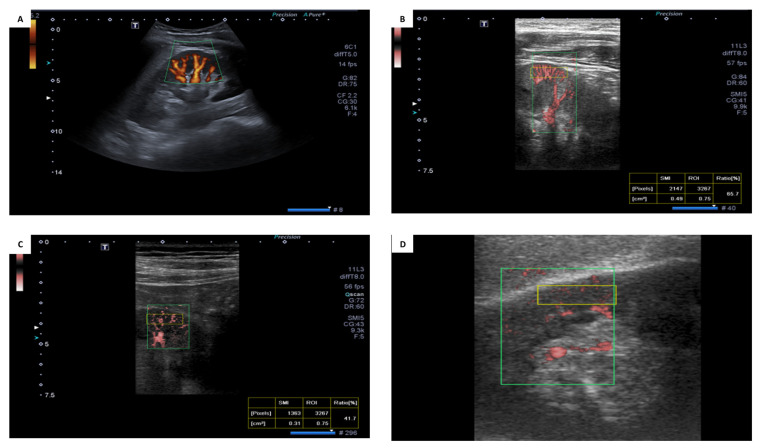
(**A**) Anatomical arcuate arteries: (**B**) SMI in otherwise healthy control; (**C**) SMI Doppler-CKD 3a; (**D**) SMI Doppler CKD Stage 3b. The analyzed fields are labeled with boxes.

**Table 1 jcm-11-00925-t001:** Description of the general knowledge score. *n* = sample size; C = control; Pts = patients; Nor = normal; SD = distorted; VD = very distorted; G = good; INT = intermediate; Po = poor.

		*n*	%
Group	C	17	24.6%
	Pts	52	75.4%
Gender	F	33	47.8%
	M	36	52.2%
Stage	1	11	21.6%
	2	13	25.5%
	3a	11	21.6%
	3b	12	23.5%
	4	3	5.9%
	5	1	2.0%
Impression	G	44	63.8%
	INT	16	23.2%
	Po	9	13.0%
SHAPE	Nor	44	64.7%
	SD	17	25.0%
	VD	7	10.3%

**Table 2 jcm-11-00925-t002:** Values of needed parameters of all subjects. Hb = hemoglobin; eGFR = estimated glomerular filtration rate; Sr. Creatinine = serum creatinine; MDRD = modification of diet and renal disease equation; CCT = creatinine clearance test.

	Mean	Standard Deviation	Percentile 25	Median	Percentile 75	Minimum	Maximum
Age	52.21	18.37	34.00	56.00	67.00	20.00	85.00
Hb (gr/dL)	13.21	1.70	11.90	13.10	14.50	9.30	17.00
Sr. Creatinine	1.22	0.57	0.80	1.13	1.49	0.49	3.50
eGFR-MDRD	73.75	34.49	43.93	64.12	105.90	14.07	158.51
eGFR-EPI	73.29	34.43	43.30	65.50	107.00	13.30	133.60
CCT	70.26	45.22	45.00	61.00	99.00	6.00	177.00
Proteinuria	1.74	1.55	0.44	1.54	2.30	0.09	6.60
Kidney size	11.19	1.52	10.30	11.20	12.00	8.30	17.20
Kidney Index	55.39	18.50	46.50	55.00	70.00	20.00	90.00

**Table 3 jcm-11-00925-t003:** Comparison between control and CKD subjects in main parameters of kidney function and SMI. C= control; P= patients; Hb = hemoglobin; eGFR = estimated glomerular filtration rate; SD = standard deviation; Sr. Creatinine = serum creatinine; MDRD= modification of diet and renal disease equation; CCT = creatinine clearance test; SMI = superb microvascular imaging.

	Group				Mann Whitney U (Z, *p*.Value)
	C (*n* = 17)		P (*n* = 52)		
	Mean ± SD	Median ± IQR	Mean ± SD	Median ± IQR	
Age	36.13 ± 12.32	33.00 + [26.00,49.00]	56.85 ± 17.24	59.00 + [45.00,72.50]	Z = −3.90, *p* < 0.01
Hb (gr/dL)	14.29 ± 1.15	14.45 + [13.50,15.30]	12.88 ± 1.72	12.80 + [11.60,13.90]	Z = −2.87, *p* < 0.01
Sr. Creatinine (mg%)	0.78 ± 0.11	0.80 + [0.74,0.84]	1.33 ± 0.59	1.27 + [0.88,1.62]	Z = −3.64, *p* < 0.01
eGFR-MDRD (mL/min)	113.44 ± 16.59	110.40 + [105.90,117.61]	63.06 ± 29.95	55.04 + [41.01,85.79]	Z = −4.58, *p* < 0.01
eGFR-EPI (mL/min)	108.32 ± 21.66	115.00 + [99.50,119.10]	62.79 ± 30.43	53.60 + [40.50,93.30]	Z = −4.24, *p* < 0.01
Creatinine Clearance (CCT) (mL/min)			70.26 ± 45.22	61.00 + [45.00,99.00]	
Proteinuria (gr/d)			1.74 ± 1.55	1.54 + [0.44,2.30]	
Kidney Size (cm)	11.19 ± 0.90	11.20 + [11.00,11.80]	11.19 ± 1.69	11.00 + [10.05,12.00]	Z = −0.47, *p* = 0.64
SMI Index (%)	72.16 ± 12.92	74.00 + [60.00,82.00]	49.91 ± 16.72	50.00 + [33.75,61.90]	Z = −4.41, *p* < 0.01

**Table 4 jcm-11-00925-t004:** Linear correlation between all main tested parameters in all participants. eGFR = estimated glomerular filtration rate; MDRD = modification of diet and renal disease equation; CCT = creatinine clearance test.

	Sr Creatinine	eGFR-MDRD	eGFR-EPI	CCT	Proteinuria	Kidney Size	Impression	Shape	Index
**Sr Creatinine**		−0.939	−0.904	−0.658	0.693	−0.123	0.645	0.646	−0.601
**eGFR-MDRD**	−0.939		0.966	0.705	−0.571	0.143	−0.678	−0.671	0.631
**eGFR-EPI**	−0.904	0.966		0.692	−0.558	0.199	−0.656	−0.644	0.593
**Creatinine Clearance (CCT)**	−0.658	0.705	0.692		−0.411	−0.017	−0.553	−0.519	0.434
**Proteinuria**	0.693	−0.571	−0.558	−0.411		−0.381	0.309	0.455	−0.251
**Kidney Size**	−0.123	0.143	0.199	−0.017	−0.381		−0.180	−0.208	0.148
**Impression**	0.645	−0.678	−0.656	−0.553	0.309	−0.180		0.996	−0.827
**Shape**	0.646	−0.671	−0.644	−0.519	0.455	−0.208	0.996		−0.815
**Kidney Index**	−0.601	0.631	0.593	0.434	−0.251	0.148	−0.827	−0.815	

**Table 5 jcm-11-00925-t005:** Linear correlation between all main tested parameters in CKD patients. eGFR = estimated glomerular filtration rate; MDRD = modification of diet and renal disease equation; CCT = creatinine clearance test.

	Sr Creatinine	eGFR-MDRD	eGFR-EPI	CCT	Proteinuria	Kidney Size	Impression	Shape	Index
**Serum Creatinine**		−0.953	−0.939	−0.658	0.693	−0.203	0.574	0.582	−0.540
**eGFR-MDRD**	−0.953		0.980	0.705	−0.571	0.212	−0.621	−0.613	0.560
**eGFR-EPI**	−0.939	0.980		0.692	−0.558	0.280	−0.642	−0.627	0.564
**CCT**	−0.658	0.705	0.692		−0.411	−0.017	−0.553	−0.519	0.434
**Proteinuria**	0.693	−0.571	−0.558	−0.411		−0.381	0.309	0.455	−0.251
**Kidney** **Size**	−0.203	0.212	0.280	−0.017	−0.381		−0.184	−0.228	0.144
**Impression**	0.574	−0.621	−0.642	−0.553	0.309	−0.184		0.992	−0.870
**Shape**	0.582	−0.613	−0.627	−0.519	0.455	−0.228	0.992		−0.853
**Index**	−0.540	0.560	0.564	0.434	−0.251	0.144	−0.870	−0.853	

## Data Availability

The data presented in this study are available on request from the corresponding author.

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
