# Peer review of "An Innovative Ultrasound Technique for Early Detection of Kidney Dysfunction: Superb Microvascular Imaging as a Reference Standard"

_jcm, 2022, doi:10.3390/jcm11040925_

Round 1
Reviewer 1 Report
In this manuscript, Armaly et al describe a study in which they use an ultrasound technique called Superb Micro-vascular Imaging to detect fibrosis in the kidney. They recruit patients with CKD and healthy controls and compare biochemical parameters from blood and urine testing to imaging parameters generated from SMI. In a subset of patients they have histopathology available as well.
Overall, I find this topic of interest, although at its stage of development, may not be of as great of interest to the general practitioner. However, the findings are compelling enough, using a potentially relatively accessible and non-invasive technique, that I have a general favorable impression of this manuscript.
However, there are numerous issues which need to be clarified and adjusted in order for me to feel confident in recommending it for publication.
- The field of nephrology is moving toward use of "kidney" over "renal" as this is a more accessible term for patients. Please change "renal" to "kidney" where possible.
- Page 2, line 72. The word "few" should be removed.
- Page 3, section 2.2. Because SMI is not a commonplace technique, the terms "impression" and "shape" used in this context are not intuitively identified as imaging parameters. The language used in the last 2 paragraphs of this section should be clarified so that it is clear you are defining these parameters.
- Along those lines, the terms "vascular index," "kidney index," "SMI index" and "index" alone are used throughout the body/tables. Are these terms all referring to the same parameter. And what is the definition of "index"? This needs to be spelled out clearly in the section describing SMI.
- There should be more detail on the imaging methods. For example, were both kidneys imaged and measurements averaged between the 2 or just one kidney? If both, were imaging parameters compared between the 2 sides? Was analysis done using ROIs? How were the ROIs made? Were the ROIs done by multiple people and with what level of training?
- There is no description of how the histopathologic findings were quantified.
- I appreciate that the patient population is weighted towards earlier stages of CKD because this is where I see the greatest potential benefit from this modality. However, CKD stages 1 and 2 are not straightforward to diagnose since GFR estimating equations are not as accurate for stages 1 and 2. There should be some clarification on how patients were categorized into the stages, particularly stages 1 and 2. Were multiple estimating equations used? Measured creatinine clearance? For stage 1 and 2, was the diagnosis confirmed on multiple measurements?
- More of a comment: It is interesting that in the control group 25% had a vascular index <60% when normal is >68%. Would you say that is just due to imprecision of the test or potential detection of mild kidney disease in the control group?
- Throughout the manuscript (title, abstract, introduction, results and discussion), there are statements that make it seem that the main finding of this study is that this technique correlates well with fibrosis detected on histopathology, when in fact this seems more of a secondary finding to me, with only a small subset (n=11) even having histopathology available and no description of quantification of histopathology. In fact, it is listed as a limitation.
- The strongest finding to me seems to be the differences in imaging parameters between healthy and CKD. Then there are modest correlations when looking at CKD population with blood parameters correlating well with each other (expected), radiologic parameters, except size, correlating well with each other (not surprising, but new); but only modest correlations between these two broader categories (blood and imaging). Urine parameters found not to correlate well. To me, this is the point that should be emphasized and described. The modest correlations are meaningful in a technique that is new and has not been explored in this arena. This fact should be emphasized. Also suggestions should be provided as to why correlations are not stronger and thoughts on potential ways to improve on imaging technique that may improve correlations.
- In the discussion, lines 339-343, these comments seem out of the blue and are not contextualized as these imaging modalities (CDUS and PDUS) were not part of the imaging analysis. Either include in results or take it out of the discussion.
- The limitation section is very brief and only lists the limitations with no description of why the limitations were necessary and suggestions for how they can be overcome in the future. In addition, some of my issues above may bring up additional limitations that should be detailed.
- In the discussion, lines 352-356 are very unclear. I read it multiple times and am not making a connection from what was in the body of the manuscript to this statement.
- In the conclusion, again there is too much emphasis of biopsy. The first conclusion that this tool could replace kidney biopsy is much too strong. Nothing in this work suggests that this technique could diagnose glomerulonephritis or interstitial nephritis or membranous nephropathy. The conclusion should focus more on the early detection aspect as its strength as this is the area where I believe this kind of a relatively non-invasive technique has real power. As far as biopsy, if diagnosis is already known, either from prior biopsy or clear biochemical findings, if this technique is eventually shown to correlate highly with degree of fibrosis, some biopsies could be avoided due to high degree of fibrosis and low likelihood of biopsy showing active disease that can be treated. But this is the less impactful finding from your studies and still requires a lot of work before it can even be considered for this purpose.
Author Response
- The field of nephrology is moving toward use of "kidney" over "renal" as this is a more accessible term for patients. Please change "renal" to "kidney" where possible.
Response: Done
- Page 2, line 72. The word "few" should be removed.
Response: Done
- Page 3, section 2.2. Because SMI is not a commonplace technique, the terms "impression" and "shape" used in this context are not intuitively identified as imaging parameters. The language used in the last 2 paragraphs of this section should be clarified so that it is clear you are defining these parameters.
Response: This statement was revised to clarify that "impression" and "shape" used in this context are not imaging parameters, rather radiologist estimation.
- Along those lines, the terms "vascular index," "kidney index," "SMI index" and "index" alone are used throughout the body/tables. Are these terms all referring to the same parameter? And what is the definition of "index"? This needs to be spelled out clearly in the section describing SMI.
Response: Yes all these terms are the same. To avoid using multiple terms, all were unified and the term SMI was used through the revised MS.
- There should be more detail on the imaging methods. For example, were both kidneys imaged and measurements averaged between the 2 or just one kidney? If both, were imaging parameters compared between the 2 sides? Was analysis done using ROIs? How were the ROIs made? Were the ROIs done by multiple people and with what level of training?
Response: One kidney was imaged and the imaging was performed by the same operator who has more than 20 years of experience (Dr. Suheli Artul) through throughout the whole study.
- There is no description of how the histopathologic findings were quantified.
Response: The term "histological changes" refers to kidney fibrosis. To avoid misunderstanding, we used in the revised MS the words "degree of fibrosis" rather than "histological changes".
- I appreciate that the patient population is weighted towards earlier stages of CKD because this is where I see the greatest potential benefit from this modality. However, CKD stages 1 and 2 are not straightforward to diagnose since GFR estimating equations are not as accurate for stages 1 and 2. There should be some clarification on how patients were categorized into the stages, particularly stages 1 and 2. Were multiple estimating equations used? Measured creatinine clearance? For stage 1 and 2, was the diagnosis confirmed on multiple measurements?
Response: We agree the reviewer that SMI could be more appreciated for early CKD stages, namely stage 1 and 2. Due to limited accuracy of MDRD and EPI-GFR equations, therefore we applied when feasible creatinine clearance test (CCT) measurement to overcome this pitfall. Since both MDRD and EPI-GFR yield comparable results we applied both of them as specified in the MS/Tables.
- More of a comment: It is interesting that in the control group 25% had a vascular index <60% when normal is >68%. Would you say that is just due to imprecision of the test or potential detection of mild kidney disease in the control group?
Response: We do not know the reason for this low percentage of healthy control subjects who have SMI of less than 60%, but we assume that it could stem from mixture of both reasons. As indicated normal health controls were choosen according to their SCr and GFR, a method that has its own limitations. Histological or vascular changes in the kidney may occur even in the presence of normal SCr or GFR. We added that to the study limitations.
- Throughout the manuscript (title, abstract, introduction, results and discussion), there are statements that make it seem that the main finding of this study is that this technique correlates well with fibrosis detected on histopathology, when in fact this seems more of a secondary finding to me, with only a small subset (n=11) even having histopathology available and no description of quantification of histopathology. In fact, it is listed as a limitation.
Response: We agree with the reviewer. In the revised MS, we emphasized the main finding which actually was not the correlation between SMI and fibrosis, rather it was the correlation between SMI and kidney dysfunction.
- The strongest finding to me seems to be the differences in imaging parameters between healthy and CKD. Then there are modest correlations when looking at CKD population with blood parameters correlating well with each other (expected), radiologic parameters, except size, correlating well with each other (not surprising, but new); but only modest correlations between these two broader categories (blood and imaging). Urine parameters found not to correlate well. To me, this is the point that should be emphasized and described. The modest correlations are meaningful in a technique that is new and has not been explored in this arena. This fact should be emphasized. Also suggestions should be provided as to why correlations are not stronger and thoughts on potential ways to improve on imaging technique that may improve correlations.
Response: Thank you for this valuable comment and your understanding of this US technology. We emphasized these points in the revised MS.
- In the discussion, lines 339-343, these comments seem out of the blue and are not contextualized as these imaging modalities (CDUS and PDUS) were not part of the imaging analysis. Either include in results or take it out of the discussion.
Response: Omitted.
- The limitation section is very brief and only lists the limitations with no description of why the limitations were necessary and suggestions for how they can be overcome in the future. In addition, some of my issues above may bring up additional limitations that should be detailed.
Response: We updated the limitations and included additional ones brought by the reviewer.
- In the discussion, lines 352-356 are very unclear. I read it multiple times and am not making a connection from what was in the body of the manuscript to this statement.
Response: This sentence was omitted as no need for it.
- In the conclusion, again there is too much emphasis of biopsy. The first conclusion that this tool could replace kidney biopsy is much too strong. Nothing in this work suggests that this technique could diagnose glomerulonephritis or interstitial nephritis or membranous nephropathy. The conclusion should focus more on the early detection aspect as its strength as this is the area where I believe this kind of a relatively non-invasive technique has real power. As far as biopsy, if diagnosis is already known, either from prior biopsy or clear biochemical findings, if this technique is eventually shown to correlate highly with degree of fibrosis, some biopsies could be avoided due to high degree of fibrosis and low likelihood of biopsy showing active disease that can be treated. But this is the less impactful finding from your studies and still requires a lot of work before it can even be considered for this purpose.
Response: We rephrased the conclusions in the discussion as well as the abstract. We emphasized the distinct differences in imaging parameters between healthy and CKD and the correlations between SMI and kidney fibrosis as secondary finding.
Reviewer 2 Report
Overall comments to the author:
In this article, Armaly Z, et al. describe the performance of Superb Microvascular Imaging (SMI) in detecting kidney disfunction and shed some light on possible aplications in the assessment of kidney fibrosis. The article has clinical relevance and originality, since it gives important information on a non-invasive technique that potentally could replace kidney biopsy in some particular scenarios.
On the other hand, major changes are needed in order to clarify some aspects and expand some relevant points.
Major comments:
- The aim of the manuscript is unclear. Authors state that the aim of the study is the assessment of renal fibrosis in CKD patients at stages 2-5 and to evaluate the capabilities of SMI for early detecting kidney dysfunction and renal fibrosis in comparison to the reference standard renal biopsy ( page 2 lines 88-95 “we aim to investigate whether renal SMI imaging can serve as a practical and noninvasive tool for quantitative 89 assessment of renal fibrosis in CKD patients at stages 2-5...” “ the current study assessed the capabilities of SMI for early detecting of kidney dysfunction and renal fibrosis in comparison 93 to the reference standard renal biopsy...”)
Since not all the patients had available renal biopsy and also not all of them where CKD stage 2-5 this should be changed all over the manuscript according to the characteristics of the study.
- I would suggest the authors to explain inclusion criteria number 2 in more detail (page 3, lines 141-142). I understand that this inclusion criteria included patients with Stage 1 of CKD but with albuminuria grade 2 or 3? If this is the case, authors should clarify that criteria.
- I would suggest to exclude the patient who had not sufficient collected data to determine CDKD stage since the correlation of SMI index and renal function is not possible.
- Since the authors mention that conventional renal ultrasound is, generally, not informative for the evaluation diabetic nephropaty, authors should point out how many diabetic patients were included in the study and which were the other presumed etiologies of CKD.
- Since only 11 out of 52 patients had a kidney biopsy available, I would suggest to change the author’s conclusions as for SMI being a potential replacement of renal biopsy and a tool to detect chronic renal fibrosis (page 10, line 359), since fibrosis can only be detected by kidney biopsy. Conclusions should be more focused on kidney function.
- I suggest to change the title in the same direction.
Minor comments:
- Authors repeat all the proteinuria results in two different pages (page 5, lines 208-2015 and page 6, lines 251-256). Even if proteinuria was only determied in CKD patients, I recommend not to repeat this information and to point out in the 3.3 section that information is available in the 3.2 section. Also, the mean value of proteinuria in CKD subjectes is described differenty : 1.47±1.55gr/24h in first lines and 1.74±1.55 gr/24 h in the following lines.
Author Response
- The aim of the manuscript is unclear. Authors state that the aim of the study is the assessment of renal fibrosis in CKD patients at stages 2-5 and to evaluate the capabilities of SMI for early detecting kidney dysfunction and renal fibrosis in comparison to the reference standard renal biopsy (page 2 lines 88-95 “we aim to investigate whether renal SMI imaging can serve as a practical and noninvasive tool for quantitative 89 assessment of renal fibrosis in CKD patients at stages 2-5...” “ the current study assessed the capabilities of SMI for early detecting of kidney dysfunction and renal fibrosis in comparison 93 to the reference standard renal biopsy...”)
Since not all the patients had available renal biopsy and also not all of them where CKD stage 2-5 this should be changed all over the manuscript according to the characteristics of the study.
Response: We thank the reviewer for this valuable comment. We have sharpened the study aims and rephrased the relevant sentences in the revised MS accordingly.
- I would suggest the authors to explain inclusion criteria number 2 in more detail (page 3, lines 141-142). I understand that this inclusion criteria included patients with Stage 1 of CKD but with albuminuria grade 2 or 3? If this is the case, authors should clarify that criteria.
Response: Thank you for this valuable note. We extended inclusion criteria #2.
- I would suggest to exclude the patient who had not sufficient collected data to determine CDKD stage since the correlation of SMI index and renal function is not possible.
Response: The CKD staging was determined for all patients.
- Since the authors mention that conventional renal ultrasound is, generally, not informative for the evaluation diabetic nephropathy, authors should point out how many diabetic patients were included in the study and which were the other presumed etiologies of CKD.
Response: The various etiologies of CKD patients is included in the revised MS (Methods)
- Since only 11 out of 52 patients had a kidney biopsy available, I would suggest to change the author’s conclusions as for SMI being a potential replacement of renal biopsy and a tool to detect chronic renal fibrosis (page 10, line 359), since fibrosis can only be detected by kidney biopsy. Conclusions should be more focused on kidney function.
Response: Thank you for the note. The conclusions were rephrased according to the reviewers’ comment.
- I suggest to change the title in the same direction.
Response: The title was changed.
Minor comments:
- Authors repeat all the proteinuria results in two different pages (page 5, lines 208-2015 and page 6, lines 251-256). Even if proteinuria was only determined in CKD patients, I recommend not to repeat this information and to point out in the 3.3 section that information is available in the 3.2 section. Also, the mean value of proteinuria in CKD subjects is described differently: 1.47±1.55gr/24h in first lines and 1.74±1.55 gr/24 h in the following lines.
Response: Thank you for the note. We omitted one of the duplications in the results section. Ina addition we corrected the minor mistake of the proteinuria: 1.47±1.55gr/24h should be 1.74±1.55gr/24h.
Round 2
Reviewer 1 Report
Thank you for addressing a number of my concerns, although there are still a few that I felt were not completely addressed:
- (correlating with #1 on my original response). renal -> kidney. I found many places where renal is still present and should be changed to kidney
- (correlating with #5 on my original response). The details of methods included in your response did not make it to the document. Also, you did not describe how ROIs were drawn.
- (correlating with #7 on my original response). I am actually not sure what EPI-GFR is. I think you mean CKD-EPI equation, which is the common nomenclature for this formula.
- (correlating with #9 on my original response). In both the title and conclusion it still reads that imaging is being compared to fibrosis, which is a biopsy tissue finding, and not measures of kidney dysfunction from blood and urine.
- Additional comment: There should be keys to your tables defining the abbreviations used.
Author Response
We thank reviewer #2 once more for his/her valuable comments
Comments
1. (Correlating with #1 on my original response). renal -> kidney. I found many places where renal is still present and should be changed to kidney.
Response: Done
2. (Correlating with #5 on my original response). The details of methods included in your response did not make it to the document. Also, you did not describe how ROIs were drawn.
Response: More details were added to the method section. One kidney was imaged and the imaging was performed by the same operator who has more than 20 years of experience (Dr. Suheil Artul) through throughout the whole study. The clearest image of the region of interest (ROIs) was chosen by Dr. Artul and analyzed accordingly. The selected ROIs represent the generalized vascular obtained mapping of the kidney cortex.
3. (Correlating with #7 on my original response). I am actually not sure what EPI-GFR is. I think you mean CKD-EPI equation, which is the common nomenclature for this formula.
Response: Thank you for the note. Indeed we meant CKD-EPI. Therefore, EPI-GFR was corrected throughout the MS.
4. (Correlating with #9 on my original response). In both the title and conclusion, it still reads that imaging is being compared to fibrosis, which is a biopsy tissue finding, and not measures of kidney dysfunction from blood and urine.
Response: We agree with the reviewer. In the revised MS, we emphasized the main finding, namely the correlation between SMI and kidney dysfunction, rather than detection of renal fibrosis. Actually, correlation between SMI and kidney fibrosis was found.
5. Additional comment: There should be keys to your tables defining the abbreviations used.
Response: All abbreviations used in the tables were added in the legends.
